# Position: Humans are Missing from AI Coding Agent Research

## Abstract

Recent progress in AI coding agent research has led to rapid improvements in agents' ability to autonomously perform complex software engineering tasks, from editing large codebases to executing long-horizon development workflows. As these systems make strides, however, the primary bottleneck to practical usefulness increasingly shifts away from pure task-solving capability, and toward challenges in how users communicate with, supervise, and trust agents.

In this position paper, we argue for a reorientation from autonomous to *human-centered* coding agents: systems designed not only to complete tasks, but to collaborate effectively with people. We identify four core interaction-level dimensions that characterize the human-agent task-solving loop: task alignment, verifiability, steerability, and adaptability. Finally, we outline concrete research directions to advance these dimensions, including user-involved coding environments, comprehensive verification mechanisms, and principled measures of human-agent interaction quality.

## 1. Introduction

Recent advances in large language models (LLMs) and agent frameworks have led to rapid progress in AI coding research. Coding agents can now modify real codebases, resolve issues in complex repositories (Jimenez et al., 2024), and execute multi-step software engineering workflows (Chan et al., 2025) that were previously out of reach for automated systems. Across benchmarks and live deployments, newer models continue to outperform earlier ones with greater autonomous execution capabilities.

However, as agent autonomy increases, the primary bottleneck to practical usefulness is shifting (Chen et al., 2025b).

---

> **Position**
>
> AI coding agent research is fixated on autonomous task completion, treating benchmark difficulty as a proxy for practical value. However, as coding agents have graduated from academic papers to industry products, we hypothesize that the next meaningful breakthroughs lie *not* in what agents can do solo, but in their interplay with human developers and users.

While earlier systems were limited by their inability to generate correct code, modern agents can produce plausible solutions, as shown by the surging scores on SWE-bench verified in Figure 1. Empirical studies increasingly highlight failures not in producing executable code, but in misunderstanding user intent (Liu et al., 2023; Jiang et al., 2022), producing outputs that are difficult to interpret or verify (Kumar et al., 2025; Treude & Gerosa, 2025), and behaving in ways that are hard to control or predict over time (Mozannar et al., 2024a; Liang et al., 2024; Chen et al., 2025b). For instance, agent-produced patches can be bloated and hard to verify, hindering their practical utility. As coding agents increasingly function as general-purpose agents (Soni et al., 2025) that program to solve diverse real-world tasks beyond software engineering (Wang et al., 2025d), these interaction challenges become even more pronounced. Consequently, as agent capabilities improve, the bottleneck shifts from "can it work?" to "can humans understand, trust, and work with it?", making human-centered design essential for translating capability into practical usefulness.

This mismatch reflects a growing disconnect between agent research and real-world deployment. Much of the field has focused on advancing autonomy, measured by success rate on harder benchmarks. Yet real-world programming is rarely a one-shot activity. Instead, it unfolds through iterative interaction, partial delegation, evolving goals, and continuous human oversight. In this setting, the goal of developing coding agents is not to replace human labor, but to augment complementary human strengths such as initiative and judgment. Crucially, human-centered interaction is not a temporary workaround for immature models: it enables new forms of collaboration in which agents and users jointly discover emerging workflows. As a result, utility depends not only on whether an agent can complete a task,

[1] Anonymous Institution, Anonymous City, Anonymous Region, Anonymous Country. Correspondence to: Anonymous Author <anon.email@domain.com>.

Preliminary work. Under review by the International Conference on Machine Learning (ICML). Do not distribute.

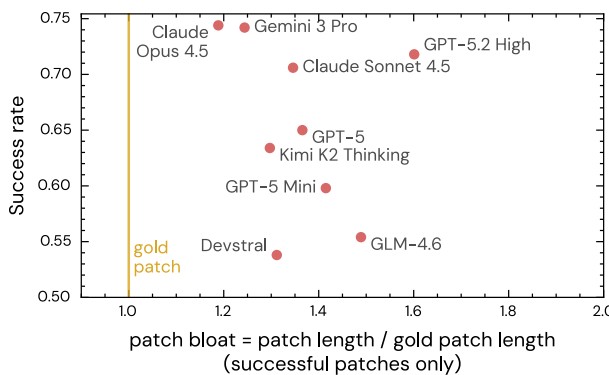

*Figure 1.* In SWE-bench Verified, despite success rate increases, agent-generated patches are consistently longer than "ideal" gold patches, posing a challenge to *verifiability* for human users.

but on whether humans can effectively work with it to shape outcomes and future work (§2).

We argue that this shift in bottlenecks calls for a corresponding adjustment in how coding agents are designed and evaluated: from maximizing agent-solo autonomy to maximizing human-centered usefulness (Position 1). To make this notion concrete, we identify four key dimensions that characterize human-centered coding agents—grounding, verifiability, steerability, and adaptability—and formalize them with measurable definitions. We choose these dimensions as interaction primitives that together span the agent task-solving cycle (Figure 2): from interpreting user intent, executing actions under human control, exposing outputs for human assessment, to improving behaviors across repeated use. Although they may not constitute an exhaustive list of desirable system properties, these foundational capabilities could support higher-level concepts such as collaboration (§3).

We also outline research directions to close this gap, including building scalable models of human users, enabling more efficient and task-aware verification, defining measurable interaction-centric evaluation signals, and exploring applications for AI coding agents beyond traditional software engineering. Together, these efforts call for new infrastructure, benchmarks, and evaluation settings that incorporate human interaction as an essential component, rather than treating it as an external afterthought (§4).

While we argue that human-centered design is essential for coding agents, this view is not without contention. We engage with common counter-arguments and clarify why these perspectives do not eliminate the need for human-centered agent research (§5). Collectively, they point toward a research agenda centered not on replacing human developers, but on building AI systems that meaningfully augment us.

## 2. AI for Code Today

Recent coding agent research has largely converged on a single objective: *increasing agent autonomy*, typically mea-

sured by an agent's ability to complete standalone software engineering tasks (Chowdhury et al., 2024; Jain et al., 2024). Under this framing, "better" agents are those that can handle harder benchmarks (Deng et al., 2025), execute longer horizons (Zhao et al., 2024a), and achieve higher end-to-end success rates. This autonomy-centric perspective has shaped several dominant research directions.

First, substantial effort has gone into *constructing more difficult benchmarks*, which demands understanding across languages (Yang et al., 2025a) and modalities (Yang et al., 2024b), execution in longer horizons (Li et al., 2024; Zhao et al., 2024a) and more complex environments (Chan et al., 2025; Yang et al., 2025b). While these benchmarks are all designed to challenge agents with increasingly complex engineering tasks, they may lead the community to overweigh tasks at the tail-end of the difficulty spectrum.

Second, to solve these increasingly difficult problems, researchers have focused on building *sophisticated agent frameworks* that scaffold LMs with crafted pipelines (Xia et al., 2024) and tools (Yang et al., 2024a; Wang et al., 2025a). Despite differences in architectures, these frameworks share a common goal of pushing agents toward fully autonomous engineering, often limiting human control, which can ultimately constrain their practical utility.

Third, the field has invested in specialized environments and data curation recipes to enable *scalable training with effective verification*. For instance, SWE-Gym established a demonstration-based training formula (Pan et al., 2025), SWE-smith and R2E-Gym automated environment scaling and verification (Yang et al., 2025a; Jain et al., 2025b), and later work diversifies task coverage (Sonwane et al., 2025; Zhu et al., 2025). These training reinforce the autonomy-centric loop, which continuously synthesizes more difficult tasks and trains agents to improve on them.

Although these research have drastically expanded agent capabilities, the assumption that continuing these efforts yields proportional practical gains remains largely unexamined. This gap stems from a mismatch between benchmarks optimized for autonomous behavior and real-world settings that inevitably involve human interaction. If we continue to prioritize isolated autonomy as the primary axis of progress, research risks overlooking how coding agents are actually used: through communication with humans, oversight and intervention from humans, and adaptation alongside humans. As a result, the field may be optimizing in a direction of diminishing returns, pursuing ever-harder tasks while marginal utility gains shrink, when the gradient toward practical impact points elsewhere: human-agent interaction.

## 3. What Users Want

Practical deployment of coding agents reveals bottlenecks that differ from those emphasized in automation-oriented

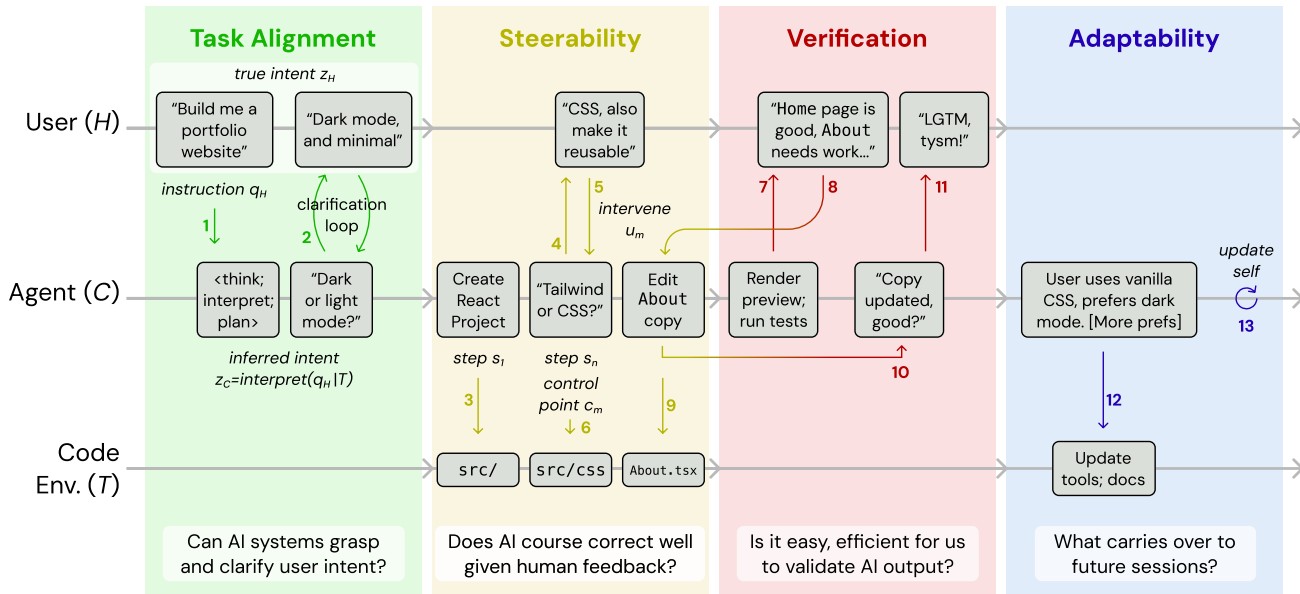

*Figure 2.* **The Human-Coding Agent Collaboration Loop.** We ground our four pillars in a unified interaction loop between a user, coding agent, and code environment. Using a concrete running example (building a personal portfolio website), we illustrate how task alignment, steerability, verification, and adaptability surface as distinct phases in a single session. The numbers denote an order in which user/agent/codebase interactions occur. Though depicted sequentially for clarity, these phases interleave in practice – task alignment questions naturally may arise during steering; verification may cause an agent to revisit intent and course-correct (step 7).

agent development. Industry reports and developer surveys from major coding tool providers consistently highlight how users are often less limited by agents' code correctness than by challenges in communicating with the agent, consuming and verifying its outputs, steering its behavior, and the agent adapting to an evolving codebase and user base.

These practical concerns motivate a shift toward human-centered coding systems design. We articulate four key pillars for building effective human-centered coding agents. In addition to qualitative characterization, we formalize these pillars with computable metrics that enable systematic evaluation and optimization. Our goal is not merely conceptual framing, but to provide actionable targets that can be reported in experiments and optimized by future systems.

### 3.1. Task Alignment

**Definition.** Task alignment refers to the process by which humans and agents establish and maintain a shared task understanding through mutual modeling (Clark & Brennan, 1991). In AI-assisted coding, users must externalize intent through NL instructions, relevant contexts, or constraints; while agents must infer latent goals, resolve underspecification, surface assumptions, and signal uncertainty. Effective alignment is therefore not a single capability, but a coordinated behavior: dynamically integrating interpretation, clarification, and revision as the task unfolds.

**Formulation.** Given an NL instruction $q_H$ from user $H$, the agent $C$ forms an internal task specification $z_C = \mathrm{interpret}_C(q_H | \mathcal{T}) \in \mathcal{Z}$ situated in environment $\mathcal{T}$ with task contexts, where $\mathcal{Z}$ denotes a structured intent space. Task alignment measures distance between the user-intended specification $z_H$ and the agent's inferred $z_C$,

$$G(H, C; q_H) = \mathrm{sim}_Z(z_H, z_C)$$

where $z_H$ can come from human annotation, and $\mathrm{sim}$ measures task similarity (cosine similarity, lexical overlap).

**Motivation.** Task alignment is key to a smooth user experience for coding agents. Industry has recognized this, producing best-practice guides on prompt engineering, instruction files, and structured frameworks for agent interaction (Ellich & Etcovitch, 2025; Cursor, 2026). Studies confirm the stakes: developers who communicate more effectively with agents see measurably better outcomes (Sarkar, 2025). When communication fails, the consequences compound. Incorrect assumptions, wasted computation, and repeated correction cycles degrade both productivity and trust (Mozannar et al., 2024a). Today, users largely adapt to agents, learning to phrase requests in ways they understand. A significant pain point is the "clarification spiral." Given a task, models launch into implementation without inferring unstated constraints and make incorrect assumptions. Users then reject results and relay additional asks, only to watch new misunderstandings emerge. What we ultimately want that remains unfulfilled is the reverse: agents that properly model users and interpret tasks.

**Research Gap.** Coding dialogue is fundamentally more demanding than traditional task-oriented dialogue. Whereas a basic request like hotel booking simply requires satisfying a fixed schema of options, programming tasks frequently necessitate diverse communication acts (Ross et al., 2023).

Despite widespread use, open conversation data between humans and AI coding systems remains starkly lacking. Datasets like WildChat and OpenAssistant capture general dialogue patterns (Köpf et al., 2023; Zhao et al., 2024b), but developer dialogue is heavily grounded in external artifacts (code, error logs, documentation) and shaped by tacit expectations around style, architecture, and implementation conventions. Today, benchmarks sidestep the study of such intricacies entirely: pass@k collapses all signal about communicative quality into a single bit, discarding everything about how mutual understanding was (or was not) achieved.

Without open coding conversation data or widely adopted open-source tooling, systematic and quantitative evaluation of grounding quality remains prohibitively out of reach (Vijayvargiya et al., 2025). Communication is instead assessed through anecdotal reports of user frustration (Jiang et al., 2022; Liu et al., 2023), leaving us without benchmarks to reliably track progress over time.

### 3.2. Steerability

**Definition.** Steerability concerns an agent's ability to expose and respond to human control signals throughout task execution. Rather than optimizing solely for uninterrupted autonomy (Horvitz, 1999), steerable agents must recognize meaningful decision points in a task, operate at appropriate levels of abstraction, and structure execution to expose decision boundaries at which human intervention can shape downstream behavior. This enables users to redirect subgoals and adjust execution strategies, while preserving agent autonomy over low-level actions.

**Formulation.** Given a user task $q_H$, the agent executes an action trajectory $\tau = (s_1, \cdots, s_N)$ where $s_i$ denotes atomic actions. The agent induces a higher-level segmentation $\mathcal{G}_C(\tau) = \{g_1, \cdots, g_M\}$, where each segment $g_m = (\tau_{k_m:k_{m+1}}, c_m)$ consists of a contiguous sub-trajectory and an associated control point $c_m \in \mathcal{C}$ (e.g., branching choices, confirmation prompts). To measure agent's responsiveness to human control, let user apply an intervention $u_m$ at the control point $c_m$, yielding a modified trajectory $\tau' = \exp_C(\tau, u_m)$. Comparing to a reference trajectory $\tau^*_{u_m}$ reflecting the intended behavioral change, we define response steerability as

$$S(H, C) = \mathbb{E}_{(m, u_m)}[\text{sim}_\tau(\tau', \tau^*_{u_m})]$$

While steerability is fundamentally defined by how agent behavior responds to human interventions, we could additionally measure structural alignment of exposed control points as a diagnostic signal. Let $\mathcal{G}^*$ denotes a reference segmentation derived from expert demonstration, we define structural steerability as $S_{\text{struct}}(C) = \text{sim}(\mathcal{G}_C(\tau), \mathcal{G}^*)$, where $\text{sim}$ measures alignment of both segmentation boundaries and control point content.

**Motivation.** Users of AI coding systems consistently express a need to control when and how it acts throughout a task (Kalliamvakou, 2025). Depending on context, users may operate at different points along the granularity spectrum (Sapkota et al., 2025): at times specifying high-level intent for agents to explore broadly via "vibe code" and rapid prototyping (Kalliamvakou, 2024; Cursor, 2026); and at other times intervening at fine-grained levels to make precise, localized changes. However, these needs cannot be met by systems that treat autonomy as a single global setting. Instead, practical steerability requires agents to identify meaningful control points in the task-solving process—moments where alternative execution paths, tradeoffs, or commitments arise—and to expose those choices to the user. By structuring execution around these control points, agents allow users to direct what happens next, rather than forcing them into a binary accept-or-reject role only after a fully specified plan has already been carried out. While task alignment governs what the user means, steerability governs what the system does next.

**Research Gap.** Steerability remains weakly supported in current coding agent systems: existing work is largely descriptive or peripheral, either documenting developer behavior shifts as agent automation increases (Chen et al., 2025b) or designing human-in-the-loop co-planning (Mozannar et al., 2025), but neither provides a formal account of how agents should structure execution to preserve user control. At present, we lack principled approaches for agents to identify and expose meaningful control points to users, leaving systems trapped between a false dichotomy between full agent autonomy and constant human supervision.

### 3.3. Verification

**Definition.** Verification refers to a user's ability to assess if a coding agent's outputs are correct with respect to task requirements. This presupposes that users can meaningfully consume and understand the agent's artifacts, including code outputs, execution traces, and intermediate reasoning (Vaithilingam et al., 2022). In practice, verification determines whether the outputs satisfy both explicit requirements and implicit constraints (Fakhoury et al., 2024). In other words, verification is about whether agent deliverables expose sufficient structure and evidence for humans to accurately judge correctness.

**Formulation.** Given a user instruction $q_H$, the agent produces output $o$, including intermediate artifacts and final output. A human verifier produces a potentially imperfect

judgment $s_H = \mathrm{verify}(o, z_H)$ due to agent output verifiability. Let $y^*(o, z_H)$ denote the ground-truth verifiers (correctness, efficiency, etc.), we define verifiability as the expected agreement between human and gold judgments:

$$V(H, C) = \mathbb{E}_{(o,z_H)}[\mathcal{A}(s_H, y^*(o, z_H))]$$

where $\mathcal{A}$ denotes an agreement metric such as accuracy.

**Motivation.** Users cannot trust what they cannot verify. Recent surveys find that of the 84% of developers now using AI coding tools, nearly half do not trust outputs while two-thirds report that half-baked solutions lead to heavier debugging burdens (Stack Overflow, 2025). Importantly, current works establish that a user's trust in coding tools is continually recalibrated through repeated verification (Sapkota et al., 2025), a process that shifts meaningfully with task stakes and user expertise (Wang et al., 2024a). For instance, while experienced developers prefer to inspect low-level implementation details, novice programmers may rely on higher-level natural language explanations and observable artifacts (Barke et al., 2023; Gu et al., 2024). Today, the burden is uneven, with less experienced developers reporting both the highest productivity gains and greatest struggle to review coding agent outputs (Sonar, 2026). Supporting verification across this spectrum is therefore critical for enabling reliable and sustained human–agent collaboration.

**Research Gap.** Unit testing is the dominant verification paradigm in current benchmarks given their precision and reproducibility. However, in practice, it captures only a narrow notion of functional correctness and often shifts the verification burden onto users. Humans rely on a much broader range of strategies to establish trust in code and the systems that generate it, such as code inspection, execution tracing, and iterative dialogue (Mozannar et al., 2024b). This mismatch suggest that while tests are valuable, they may be insufficient for supporting verification in the wild.

First, tests do not always reduce verification effort. Developers rarely construct test suites before prompting AI tools (Sonar, 2026); instead, tests are generated with code. When AI is producing both, the tests no longer serve as independent evidence for building trust. Rather, in addition to code, users must now check if tests capture intent – a lateral shift, not reduction, in cognitive load. To be clear, unit tests are valuable when constructed strategically, by encoding requirements once to avoid repeated verification. But using tests as the default mechanism for any agent output creates work that many interactions do not warrant.

Second, while unit testing has its place, it can be ill-suited for non-technical users and tasks beyond traditional software engineering. Code is increasingly the action space for general-purpose agents, powering exciting use cases such as data analysis (Gu et al., 2024), interface design (Yuan et al., 2025), and tasks on the edge of imagination (e.g., raising a plant[1]). Consequently, outputs are becoming increasingly heterogeneous and context-dependent, making universal pass/fail criteria hard to define (Barr et al., 2014).

Rather than relying solely on test-centric workflows, verification should surface evidence in human-interpretable ways, such as visual previews and interactive summaries. Verification needs to be designed in ways that reduce user burden and make agent behavior easier to assess, especially for users without deep programming expertise.

## 3.4. Adaptability

**Definition.** We define adaptability as an AI coding agent's ability to maintain and update itself using accumulated experience, to improve future performance while preserving previously acquired capabilities (Wang et al., 2024b). This includes updating persistent memory such as user preferences (Letta, 2025) and task specifications (Wang et al., 2025c), as well as action policies by acquiring and refining reusable skills (Wang et al., 2025b). Beyond storing historic context, highly adaptive agents leverage experience to improve how they perceive the environment and take actions, generalize to related tasks while avoiding undesirable drift.

**Formulation.** Let $\tau$ depict a distribution (user task, preference, etc.), on which the agent aggregates task experiences $e \sim \tau$ and updates itself $C^k = \mathrm{adapt}(C, \{e\}_1^k)$. We quantify adaptability as the improvement over task sessions

$$A(C^k) = \mathbb{E}_{\tau \sim \mathcal{T}}\left[\mathrm{Perf}(C^k) - \mathrm{Perf}(C^0)\right]$$

where $\mathrm{Perf}(\cdot)$ denotes a task performance metric. This formulation captures the agent's ability to improve through experience toward a target distribution.

**Motivation.** Software engineering is inherently adaptive: Codebases are living artifacts that developers tend to, expand, refactor, and repurpose over time. Users increasingly expect the same continuity and growth from their AI coding tools, yet a common pain point is the need to re-establish context and re-state execution details every session, often described as "prompt fatigue" (Lessard, 2025). Early efforts in building extensions that auto-generate documentation to maintain persistent memory, suggest reductions in user cognitive load and more efficient, personalized interactions (Katz, 2026). Yet adaptability extends beyond remembering contexts to upgrading their actions, by iteratively developing their skill repertoire and reusing them effectively. However, the design of such systems is still in its infancy. Today's agent "memory" and "skills" are often little more than indexed markdown files (Anthropics, 2025), rather than mechanisms for sustained learning, leaving the promise of genuinely adaptive coding agents largely unrealized.

---

[1]See the AutonCorp Biodome Project, where Claude autonomously manages a tomato plant via IoT sensors.

**Research gap.** Today's coding agents are developed and evaluated on isolated, one-off tasks, with no incentive for learning from prior sessions or accumulating user-specific context. Even when tasks share a repository or creator (e.g., 850 SWE-bench tasks from Django), solving one has no effect on a later task. Simply designing harder problems that require more turns to solve does not obviate the need for persistent context across sessions, a key to sustained support for human engineers. After all, beyond functional code, repositories embody collaboration across developers, accumulating a collective of conventions, preferences, and institutional knowledge over time. Systems that internalize this are most effective at reducing the cognitive burden of repeating context over multiple runs.

When adaptability is studied, it optimizes for agents, not users. Agent Skill Induction (Wang et al., 2025b) and Live-SWE-agent (Xia et al., 2025) show that agents can acquire reusable skills and improve task performance across sessions, but the metric remains squarely on task success, with no notion of whether or how a human copilot might benefit. The gaps come to light when collaboration with a human-in-the-loop extends beyond a single session. A developer tells an agent to use `logging` instead of `print`. Does the lesson persist to next week, or vanish by the morning? If contradictory preferences are introduced, can the agent sort out the conflicts and ask for confirmation if needed? As discussed in *Motivation*, practitioners are already improvising solutions (`AGENTS.md` file, custom rule sets) with promising results. But systematically evaluating whether current and future proposals improve multi-session human-agent collaboration remains uncharted.

### 3.5. Isn't there more?

Beyond these four pillars, readers may naturally wonder about other properties, such as safety or proactivity. As discussed in §1, we view additional directions as complementary rather than foundational. They either emerge as compositions of core dimensions (e.g., safety depends on grounding and verification working together) or reflect system-level design choices, not underlying interaction capabilities. Find discussions of several such properties in §B.1.

### 4. Closing the Gap

To tackle the gaps from §3, rather than prescribing isolated fixes per area, we consider: "What infrastructure missing today blocks progress across multiple dimensions?"

A common bottleneck across all pillars is that our formulations require sampling distributions of human behavior – intent, judgments, interventions – that current open-source pipelines do not provide at scale. Furthermore, we lack task structures that truly require interaction to succeed and de-

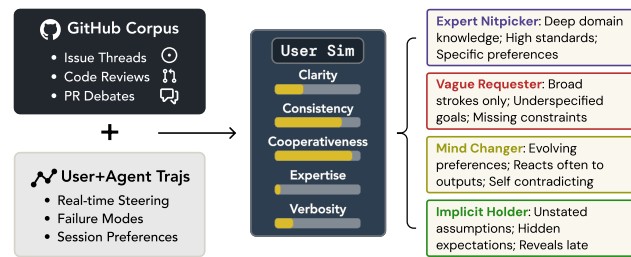

*Figure 3.* **Generating Diverse Simulated Users.** We envision combining GitHub interaction data with observed human-agent trajectories to parameterize user simulators along dimensions like clarity, consistency, and expertise.

mand verification approaches beyond unit tests, particularly in application domains beyond software engineering.

We identify four high-leverage research directions, each targeting a bottleneck that stalls multiple fronts simultaneously. For each, we discuss how progress is being throttled, then propose potential solutions.

### 4.1. Scale human modeling

Today, researchers face an unfortunate dilemma when evaluating human-AI collaboration: either run expensive human studies that do not scale, or default to fully autonomous benchmarks that sidestep the human entirely. The machine learning community has largely chosen the latter. But for genuine progress on human-centered coding agents, there is no shortcut. The formulations in §3 all require sampling from human distributions: specification ($\mathcal{G}$) necessitates user intent $z_H$, verification ($\mathcal{V}$) depends on human judgments $s_H$, and steerability ($\mathcal{S}$) relies on human intervention $u_m$.

To address this, the first path is to construct *user simulators* that faithfully model humans as active software developers and users (Yao et al., 2024). A key challenge is that current simulators prompted to "act like a human" behave homogeneously and overly cooperative (Li et al., 2017). More prompting alone is not sufficient: faithful simulation requires modeling writing style, personal preferences, expertise, and realistic failure modes, since real users often struggle to articulate constraints, change their minds mid-task, and hold implicit expectations they never verbalize (Naous et al., 2025; Shi et al., 2025). One promising approach is to leverage large-scale developer profiles and interactions available on GitHub (Figure 3), such as pull request histories, code review discussions, even real user-agent interaction data, which together capture user behavioral and preferential patterns to train more realistic interlocutors.

A complementary approach is to build widely adopted user-facing platforms that *collect interaction data as a byproduct of real use*. For instance, Copilot Arena integrates into IDEs and gathers large-scale preference signals on daily coding tasks that substantially diverge from model rankings

*Figure 4.* **Context should determine the "shape" of oversight.** Rather than defaulting to unit tests or code diffs, AI systems should surface artifacts that make validation intuitive.

from static benchmarks (Chi et al., 2025). We encourage researchers and industry to partner and share anonymized interaction data to enable scalable research.

Ultimately, measuring and improving the fidelity of LM-based simulators remains an open challenge. How do we evaluate alignment to user intent and behavior, ensure diversity without collapsing to stereotypes, and effectively combine simulated and real interaction signals?

### 4.2. Enable efficient oversight

As discussed in §3.2 and §3.3, today's users verify agent outputs in tedious and manual ways. Proxies of code correctness, such as unit tests, scale poorly in the wild and may not be well-suited to the task or end user's expertise.

Rather than placing the full burden on users, coding systems should proactively support oversight (Chen et al., 2025c; Sun et al., 2025; Zhao et al., 2025; Raghavendra et al., 2026). Recent work has explored generate relevant tests without explicit requests (Chen et al., 2022; Mündler et al., 2024; Jain et al., 2025a) and self-checking outputs prior to delivery (Chen et al., 2023; Ni et al., 2023).

Supplanting such traditional methods, we envision verification becoming a dynamic, protean procedure. Conditioned on task and user, agents should reason about notions of quality and surface appropriate artifacts to make validation tractable, as imagined in Figure 4. A data pipeline implementation might warrant a `.md` file summarizing key steps (e.g., how were missing values handled). On the other hand, code for a webpage should be presented visually for humans to assess design quality at a glance.

Several open questions remain. How should we evaluate the quality of verification approaches themselves? What communicative cost can users bear before cognitive load degrades their judgment? And when should agents self-verify versus surface as artifacts for human review?

### 4.3. Define measures for interaction

The current focus on task resolution leaves interaction quality unmeasured and unoptimized. §3's formulations all involve expectations over human behavior, such as intent ($z_H$), judgments ($s_H$), or interventions ($u_M$). We hypothesize that operationalizing such latent variables has value.

How many turns does it take to recover $z_H$? How often is $u_M$ required? How much human effort does $s_H$ demand?

Fortunately, we need not start from scratch. Decades of HCI and software engineering research offer rich precedent for defining and operationalizing interaction quality from user studies (Sarkar et al., 2022; Sun et al., 2022; Sergeyuk et al., 2024). Mozannar et al. (2024a) introduces the CUPS taxonomy, a categorization of programmer-AI interaction into 12 discrete states (e.g., "Prompt Crafting", "Editing Recent Suggestions"), which they then use to measure the distribution of time and transitions across these states. Barke et al. (2023) distinguishes acceleration mode (user knows what to do, measures time-to-completion) from exploration mode (user uncertain, measures validation effort and suggestion-browsing depth). Chen et al. (2025a) proposes the PULSE framework to operationalize user satisfaction as a training signal, using models to predict satisfaction from interaction traces and project the effect of agent design choices. Literature in such fields is ripe with such gems; model trainers and agent builders could port and operationalize them at scale.

Beyond adapting existing frameworks, there is gold to be found in large-scale trajectory analysis. Cursor's Tab RL work (Jackson et al., 2025) illustrates this: by mining 400M+ daily accept/reject decisions, they discovered that *when* to suggest matters as much as *what* to suggest. This insight is invisible to pass-rate benchmarks. Infrastructure for agent trajectory analysis (Bouzenia & Pradel, 2025; Dunlap, 2025; Meng et al., 2025) presents an exciting opportunity to rapidly define and validate human-centered interaction metrics like collaboration quality and user satisfaction.

### 4.4. Go beyond software engineering

Coding agents are general agents (Soni et al., 2025). Any task expressible programmatically becomes fair game. Manage a stock portfolio. Control a smart home. Tend a greenhouse. Plan a wedding. Compose music. People are already using coding agents for these and more. This shift demands that agent design move beyond tools optimized for professional developers to support everyday users.

What makes such domains compelling is that the four pillars arise intrinsically. Grounding user intent is essential: when an agent managing a portfolio is told "be more conservative", this means different things from a retiree versus a college graduate. Orchestrating a smart home foregrounds verification; if "unlock the front door" or "turn on the stove" leads to mistakes, the damage can't be undone with a simple `git revert`. Steerability is indispensable for planning tasks, where flights get canceled, open slots get booked, and user preferences change. Consider monitoring a greenhouse. The right watering schedule depends on this morning's humidity, but optimal planting strategy is best informed by last season's growing cycles. An agent must digest feedback sig-

nals spanning hours to months, deciding which timescales to attend for each decision. This is adaptability at its purest.

## 5. Alternative Views

We dedicate the following section to addressing meaningful points of contention to arguments put forth by our position.

**(a)** *Given the rapid rate of progress, AI will take over coding. Studying how humans interact and collaborate with AI coding systems is a fleeting need.*

Anthropic CEO Dario Amodei famously stated in March 2025 that in the near future, AI will write 90% of code (CFR, 2025), a declaration representative of a general sentiment that ongoing improvements in models' coding capabilities will eventually stamp out any need for human effort in software development (Simon, 1965; Collins, 2024).

We posit that whether or not this future materializes, the study of how humans participate and benefit from an AI-enhanced coding loop will remain relevant. Even if Dr. Amodei's prediction bears out, humans do not exit the loop. As agents carry out increasingly complex tasks, more decisions must be made, many of which are personalized, organization-specific, or depend on information only a specific individual possesses. The locus of effort simply shifts upstream, from implementation to specification and evaluation. Code, after all, is simply the language we use to communicate with machines – one that continues to evolve, from assembly to C to Python. Assuming agents bear more responsibility for writing Python, developers may instead convey intent with natural language or perhaps a new formalism balancing human expressiveness with programmatic precision. But the task of coding, fundamentally how we communicate with machines, does not vanish; it transforms.

Present-day trends already reflect this. Concurrent with the emergence of more end-to-end coding agents (Anthropic, 2025; OpenAI, 2025), tools designed to aid developers, not replace them, have proliferated: for reviewing (Graphite, 2025; Greptile, 2025), editing front-ends (Ginsberg & Lu, 2025), completion (GitHub), and debugging (Levin et al., 2024; Zhou et al., 2025a). The spectrum of tools coming to market suggests that users not only prefer copilots in some cases, but also desire different interfaces for different tasks. Both in-IDE auto-complete and a visual editor can be used to modify webpage elements, but which a developer reaches for depends on the task, their expertise, and preferences. Beyond tools, the outcomes also embed personal taste. Should a button be red or blue? Styled with inline CSS or reusable classes? Accessed via a modal or new page?

**(b)** *Human interaction and evaluation are too costly to scale.*

Human evaluation for natural language generation systems is challenging (Celikyilmaz et al., 2020). Studies are expensive and time-consuming, particularly for tasks requiring heavy domain expertise like coding (Howcroft et al., 2020). A lack of standards around how to design, execute, and report human evaluations can easily lead to irreproducible results (Van der Lee et al., 2021).

These concerns, while valid, can be overcome. As covered in §4, just as the NLP community developed scalable proxies of human feedback to improve instruction following and alignment (e.g., reward models, LM-as-judge (Dubois et al., 2023; Zheng et al., 2023)), metrics such as cyclomatic complexity (McCabe, 1976), cognitive complexity (Campbell, 2018), maintainability index (Oman & Hagemeister, 1992) and readability (Buse & Weimer, 2008; 2009) are computable and tangibly alleviate maintenance burden. More ambitiously, recent works have showcased the viability of benchmarks with user simulators (Yao et al., 2024) and preference collection through organic usage (Chiang et al., 2024) as paths towards scalable human-centered evaluation. The challenge of standardizing such nascent approaches is precisely why now is the time to act (Shao et al., 2025b).

**(c)** *Capabilities first. Human-centered concerns are product problems that model builders shouldn't get distracted by.*

The Bitter Lesson argues that general methods leveraging computation ultimately supersede manually crafted approaches based on human knowledge of a domain (Sutton, 2019). If anything, the current market reflects this division of labor. Researchers optimize capabilities, then product builders graft the appropriate user interfaces downstream.

Our position is not anti-scaling. Rather, we agree, with the important caveat that what we're actually scaling toward matters. If we continue optimizing solely for autonomous coding agents, we will produce just that. Better collaborators will not emerge for free (Shao et al., 2025a; Shen et al., 2025; Weston & Foerster, 2025; Wu et al., 2025; Zhou et al., 2025b). The leap from GPT-3 to ChatGPT illustrates this point. Incorporating human preferences into the training objective itself transformed a research artifact into a gripping conversationalist (Ouyang et al., 2022). Deferring human-centered concerns to downstream product work risks entrenching interaction patterns that are difficult to undo.

## 6. Conclusion

We contend that the omission of consideration for humans in AI coding agent research threatens to undermine the very utility these systems seek to provide. Left unchecked, the gap between AI coding agent research and real-world utility may very well widen. The research community must decide whether to optimize for leaderboards or the people who actually use these systems as well. Otherwise, we run the risk of building ever-more-capable coding agents that fewer people know how to wield.

## Impact Statements

We highlight two potential impacts of this position paper on the research community and broader society.

First, our work aims to point out a potential symbiosis between HCI research and agent development in the ML community. While HCI studies how people interact with AI coding systems, much agent research continues to prioritize autonomous task completion. By introducing measurable human-centered objectives, we provide a shared framework that can foster closer integration across ML, HCI, software engineering, and related fields.

Second, our position reframes automation in software work from maximizing autonomy to supporting meaningful human interaction. Rather than replacing programmers, human-centered coding agents can amplify human judgment and initiative while remaining controllable and interpretable. By advocating for accessible interaction and verification mechanisms, this approach also broadens access to programming capabilities for non-experts, enabling wider participation in AI-powered tools. More in §B.2.

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
