# A. Metrics

In this section, we review our methodology for computing "human-centered" metrics related to code quality that we used to create the graphs in the main paper. We also include additional plots and analysis discussing further findings in our investigation of how well code meets standards beyond functional correctness.

## A.1. Patch bloat

Our patch bloat metrics quantify the size of the changes that the LM applied (*submitted patch*) relative to the human solution (*gold patch*). Consistently large patches are not only a problem for code review and *verifiability*, but can also point to over-engineered or overly complex solutions (again hampering verifiability) or *task grounding* issues.

We compute these metrics for models evaluated on the main SWE-bench leaderboard (SWE-bench Verified, using mini-swe-agent). To avoid confounding effects, we restrict analysis to successfully resolved task instances.

First, we clean both the submitted patch and the gold patch by removing

 (i) all newly added files (for example, this removes reproduction scripts and other auxiliary files from the submitted patch),

 (ii) all non-Python files (SWE-bench instances are all from Python repositories; this removes pure documentation changes and any other artifacts), and

 (iii) changes to all test files (we want to focus on the quality of the implementation and remove work on tests as a confounding factor).

We then calculate the *bloat ratio* as the ratio of character lengths between the submitted and gold patch. We show results for two metrics

 1. *Average bloat ratio:* The average of the bloat ratio across all resolved task instances.

 2. *Bloated patch fraction:* The fraction of resolved task instances where the bloat ratio exceeds 1.5.

As observed already in Figure 1, not only do LMs have a consistently high average bloat ratio, but it also is largely uncorrelated with the task solving ability of the models as measured by the task resolution rate. Figure 5 further shows that there is also a positive trend with model release date in both average bloat ratio and bloated patch fraction. All models show a bloated patch fraction of more than 15%, demonstrating that the high average bloat ratio is not only an artifact of outliers.

## A.2. Further insights on patch bloat

We annotate all submitted patches that are longer than their gold patch counterpart (i.e., that contribute to an average bloat ratio $> 1$) using GPT-5 mini and Claude Haiku 4.5 as a judge to identify the model behaviors that drive patch bloat. The results are shown in Figure 6. Both models flag *verbose implementation* (e.g., unnecessary assignment of intermediate variables) as one of the leading causes for the longer patches, affecting around 60% of bloated resolved patches. Next most prevalent is *scope creep* (50–65%) and *overly defensive* code (20–30%). Excessive documentation and generally over-engineered solutions are flagged for around 20–30% and 10% of bloating.

The following system prompt was used for annotations:

```
You are an expert software engineer analyzing why a submitted patch is LONGER than a gold (reference) patch.
Both patches SUCCESSFULLY solve the same problem, but the submitted patch has more lines of code.
Your task is to identify the reasons for this extra length.

The gold patch is assumed to be the minimal, optimal solution.

You are analyzing a submitted patch that is LONGER than the gold patch. Both patches solve the same problem.
Your task is to identify WHY the submitted patch has more lines of code.

Use ONLY these 5 categories:

## overly_defensive

The submitted patch adds defensive code that the gold patch does not include:
- Try/except blocks or error handling not present in the gold patch
```

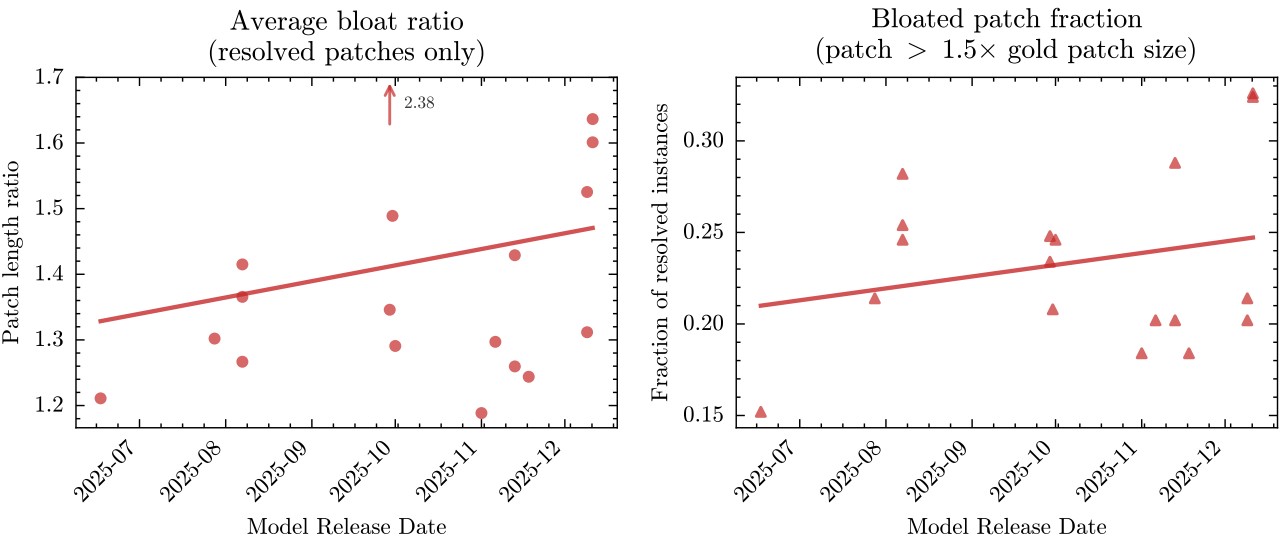

*Figure 5.* LM generated patch sizes are on the rise.

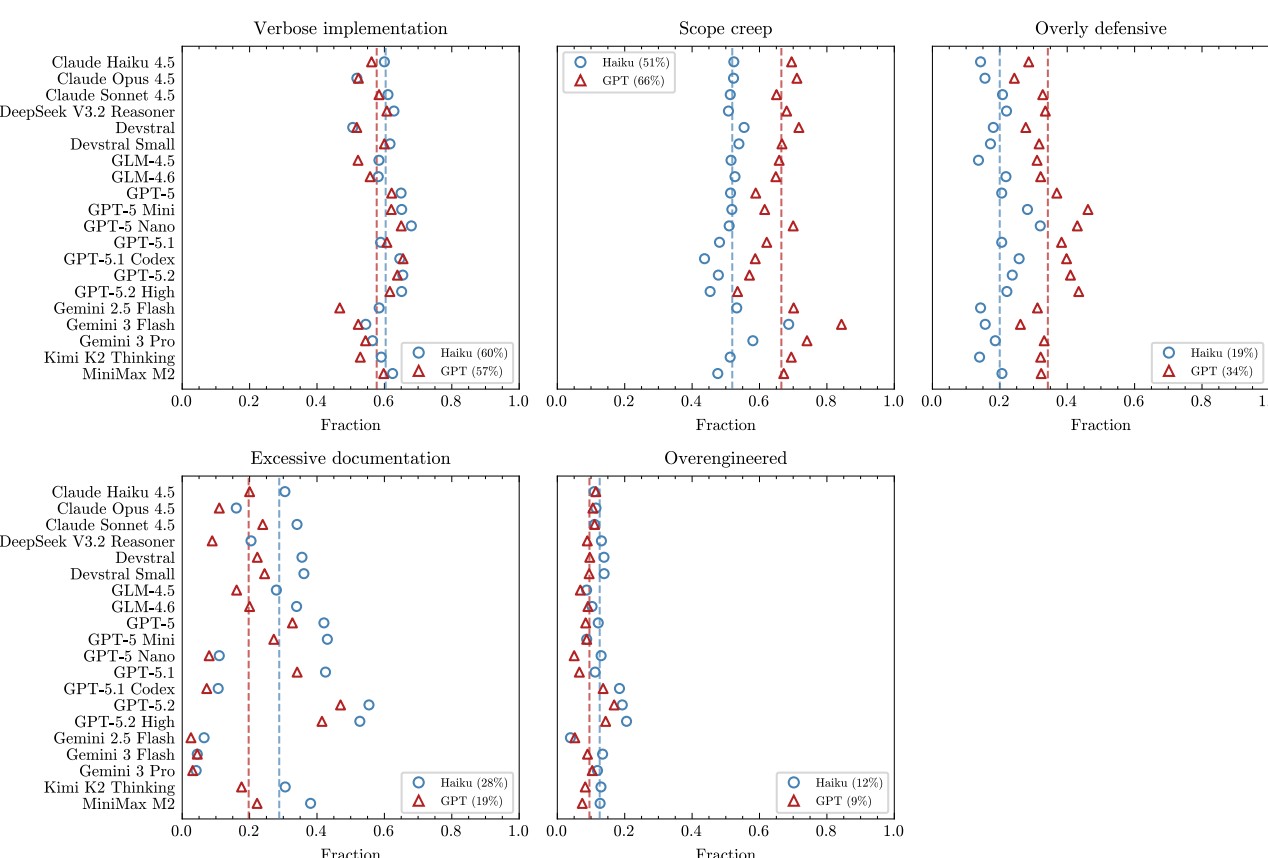

*Figure 6.* Annotation of factors that cause LM submitted patches to be longer than the human gold solution. Fraction refers to the fraction of submitted patches that resolve the task instance but are longer than their gold patch counterpart. The model average is shown as dashed lines with its numerical values quoted in the figure legends.

- Type checks, None checks, or input validation not present in the gold patch
- Broader exception catching than necessary
- Explicit error returns or fallback paths that aren't needed
- Defensive API options (e.g., errors='ignore') not in the gold patch

Use this category when the extra code is about "being safe" or handling edge cases
that the gold patch trusts won't occur.

## scope_creep

The submitted patch makes changes beyond what's needed to fix the issue:
- Modifying behavior in ways unrelated to the fix
- Adding features or functionality not requested in the problem statement
- Reformatting existing code (whitespace, quotes, import ordering, line breaks)
- Modifying dependencies and imports
- Touching files or functions that the gold patch doesn't touch
- Removing or adding comments unrelated to the fix

Use this category when the extra code comes from doing MORE than what was asked,
not from doing the same thing in a longer way.

## excessive_documentation

The submitted patch adds comments or documentation that the gold patch does not:
- Inline comments explaining obvious code
- Multi-line comment blocks describing the change
- Docstring additions or modifications not in the gold patch
- Comments that describe "what changed" rather than "why" (temporal comments)
- Redundant documentation of self-explanatory logic

Use this category when the extra lines are comments/docs, not executable code.

## verbose_implementation

The submitted patch implements the same fix but with more code than necessary:
- Writing explicit loops instead of comprehensions or built-in functions
- Duplicating logic that could be factored out or reused
- Not using existing utility functions, APIs, or methods that the gold patch uses
- Using multiple statements where one would suffice
- Overly explicit variable assignments or intermediate steps
- Longer conditional chains that could be simplified

Use this category when the submitted patch does the SAME thing as the gold patch
but uses more lines to express it. The logic is equivalent, just wordier.

## overengineered

The submitted patch introduces unnecessary abstraction or architectural complexity:
- Creating new classes, functions, or modules that the gold patch doesn't need
- Adding configuration options or parameters for flexibility that isn't required
- Implementing generic solutions when a specific fix would suffice
- Adding layers of indirection (wrappers, decorators, factories) not in the gold patch
- Building infrastructure for future extensibility that isn't asked for

Use this category when the extra code comes from ARCHITECTURAL overhead - new
abstractions, indirection, or generalization beyond what the problem requires.

For each reason you identify, provide:
- category: One of the exact category names listed above
- reason: A brief explanation of this specific instance (1-2 sentences)

You may assign multiple categories if the extra length has multiple causes.
You MUST return at least one category.

Together with the following user prompt:

```
<problem_statement>
{{ problem_statement }}
</problem_statement>

<gold_patch>
{{ gold_patch }}
</gold_patch>

<submitted_patch>
{{ submitted_patch }}
</submitted_patch>

The submitted patch is LONGER than the gold patch. Identify the categories that explain this extra length.
You MUST return at least one category.
```

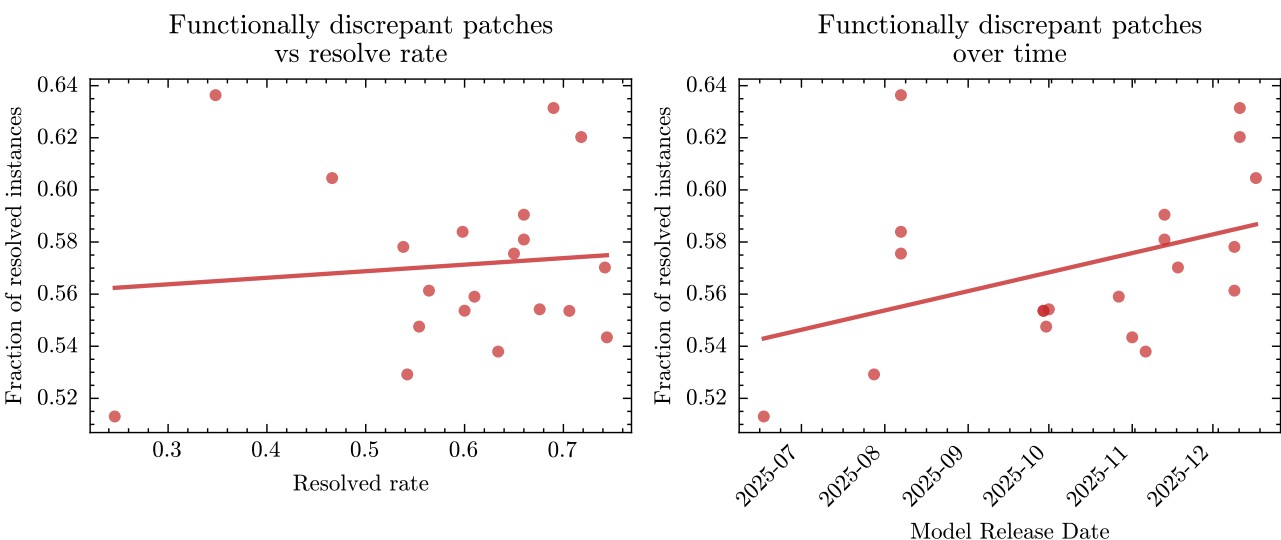

*Figure 7.* Functional discrepancies in resolved patches

## A.3. Functional differences

SWE-bench task instances are scored as resolved (1) or unresolved (0) based on whether all unit tests (including some that were not visible to the agent during inference time) are passing. However, tests rarely cover all behavior details[2]. This means that even resolved instances can have functional discrepancies with the gold patch. This does not necessarily mean that they are incorrect solutions to the problem statement, but might also be due to ambiguities or degrees of freedom of the problem statement and implementation decisions that nonetheless have significant downstream impact. Seeing functional discrepancies even in patches that pass unit tests therefore points to the need for dialogue-based *task grounding* (to resolve ambiguities and obvious missing specifications), *steerability* (to keep the user in the loop for implementation decisions that are not obvious at the beginning of a trajectory) and good *verification*.

To quantify this amount of functionally discrepant patches, we perform an annotation similar to §A.1. We perform the same cleaning of patches, and then use Claude Haiku 4.5 and GPT-5 mini as a judge to annotate the tuple of problem statement, submitted patch, and gold patch. The LM responds with structured output, returning a list of possible issue categories together with reasoning about why they apply.

To adopt the most conservative stance, we only consider a submitted patch as functionally discrepant if both models flag at least one functional discrepancy. The results are shown in Figure 7. We observe that:

1. The discrepancy is higher than 50% for all models.

2. Resolve rate is weakly correlated with an increase in functional discrepancies.

3. Functional discrepancies are more prevalent in patches submitted by more recently released models.

Details on the results of both models are shown in Figure 8. The number of instances that are flagged by either model are relatively consistent: Across all experiments, Claude Haiku 4.5 flags 66% of resolved patches as inconsistent, GPT-5 mini 62%. The fraction of patches flagged by both models at the same time is 57% (this corresponds to the numbers shown in Figure 7 and the red bars in Figure 8). Because the categories of the discrepancies cannot be separated clearly, there is some disagreement between the models when considering individual categories (for example, whether a discrepancy affects "standard behavior" or "edge case handling"). Standard and edge case behavior each affect around one third of patches. Around 20% of patches miss functionality that is included in the gold patch while more than 10% of patches include unrelated changes that aren't included in the gold patch.

---

[2]In particular, because there is a balance to be struck with tests not getting overly specific with respect to the specifications in the issue text

For the annotation we use the following system prompt:

```
You are an expert software engineer analyzing patches/diffs for code changes.
Your task is to compare a submitted patch (<submitted_patch>) with a gold (reference) patch (<gold_patch>).
The gold patch is the reference solution assumed to be correct.
Your job is to identify and categorize any functional discrepancies in the submitted patch relative to the gold patch.

## Definition of "Functional"

A "functional" change affects the observable behavior of the code, including

- return values,
- exceptions raised,
- side effects,
- I/O,
- externally-visible state changes.

Treat the gold patch as the reference even if you personally disagree with it.

The following are NOT functional changes:

- Logging or print statements
- Wording of messages that are not clearly specified in the problem statement
- Documentation or comments
- Code formatting or style
- Import ordering
- Variable/function naming (unless it changes a public API that callers rely on)
- Implementation details that produce identical behavior

Message text IS a functional discrepancy only if the problem statement explicitly specifies the exact message content or a strict
    message format.

## Categories

Use ONLY these categories:

## standard_behavior

The submitted patch produces different behavior than the gold patch for standard, typical inputs implied by the problem statement.
This excludes differences in message wording, comments, or non-functional aspects.

## edge_cases_handling

The submitted patch handles edge cases or error conditions differently than the gold patch.
Edge cases include: null/None values, empty collections, zero values, boundary values, or error paths.
Use this category when the difference is limited to exceptional/boundary conditions; if the difference affects typical inputs, use
    `standard_behavior` instead.

Examples:
- Additional error return paths not in the gold patch
- Adding options like `error='ignore'` not present in the gold patch
- Catching exceptions and suppressing them (e.g., replacing with print statements) instead of propagating
- Returning different values for boundary inputs

NOT a functional discrepancy:

- Re-raising caught exceptions with the same or equivalent error type
- Different wording of error messages

## missing_functionality

The submitted patch omits functionality that is present in the gold patch.
The gold patch implements something that the submitted patch does not implement at all.
This is different from different_behavior – here the submitted patch simply lacks the functionality entirely.
Use this when the gold introduces a new behavior/branch/check/output and the submitted does not implement it in any form.

## unrelated_changes

The submitted patch makes functional changes unrelated to the problem statement that are also absent from the gold patch.
This includes new features or enhancements beyond the scope of the problem.
This excludes non-functional changes (documentation, comments, formatting, imports).

## fundamentally_different_approach

The submitted patch solves the problem using a fundamentally different approach than the gold patch.
Example: Modifying different (non-private) functions to achieve the solution in such a way that
there are non-trivial functional discrepancies as a result.

Note: This category describes the structural approach.
If a fundamentally different approach also causes specific behavioral discrepancies
(e.g., edge case handling differences), report BOTH:
- One `fundamentally_different_approach` item for the structural difference
- Separate items for each distinct behavioral discrepancy (e.g., `edge_cases_handling`)

## Output Format

You MUST return a single JSON object with exactly this shape:
- reasons: a list of objects, each having:
- category: one of the exact category paths listed above
- reason: a brief explanation (1-2 sentences) that names the triggering condition/input and the behavioral difference (gold vs
    submitted)
```

```
Return a separate item for each distinct functional discrepancy. You may use the same category multiple times.

Example: If the submitted patch

(1) modifies different functions than the gold patch,
(2) handles None values differently,
(3) handles empty lists differently, and
(4) adds an unrelated feature,

return four items in `reasons`:

- `fundamentally_different_approach` (for #1)
- `edge_cases_handling` (for #2)
- `edge_cases_handling` (for #3)
- `unrelated_changes` (for #4)

If both patches are functionally identical, return:
{"reasons": []}
```

The user prompt is formatted with the problem statement, submitted patch and gold patch as follows:

```
<problem_statement>
<problem_statement>
{{ problem_statement }}
</problem_statement>

<gold_patch>
{{ gold_patch }}
</gold_patch>

<submitted_patch>
{{ submitted_patch }}
</submitted_patch>

Categorize any functional discrepancies in the submitted patch compared to the gold patch.
If there are no functional discrepancies, return {"reasons": []}.
</submitted_patch>

Categorize any discrepancies (if present) of the submitted patch compared to the gold patch.
It's ok to return an empty list if there are none.
```

The following structural output format is enforced:

```
CategoryType = Literal[
    "standard_behavior",
    "edge_cases_handling",
    "missing_functionality",
    "unrelated_changes",
    "fundamentally_different_approach",
]

class Reason(BaseModel):
    category: CategoryType
    reason: str

class PatchAnalysis(BaseModel):
    reasons: list[Reason]
```

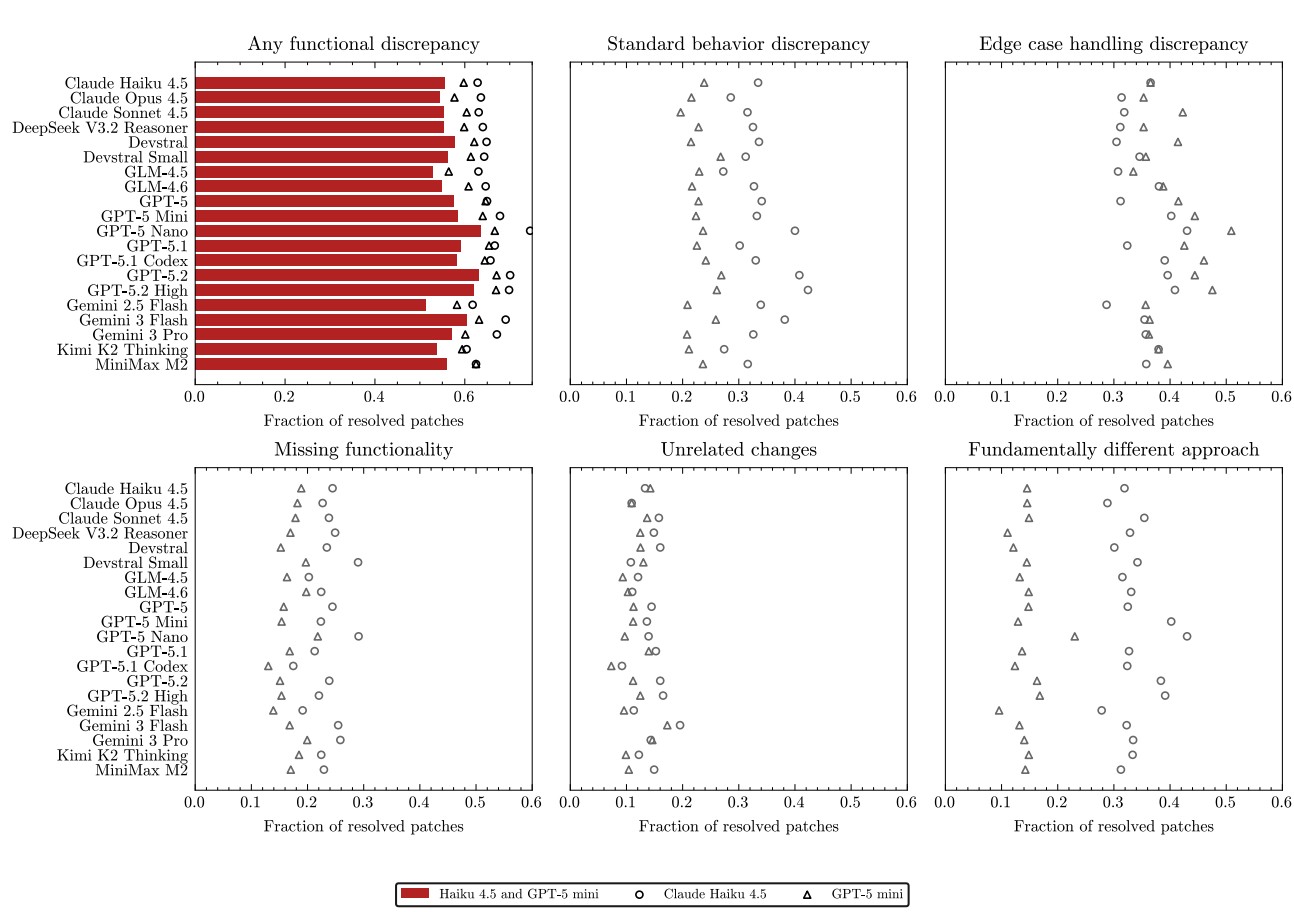

*Figure 8.* Details on the result of annotating the submitted patches for functional discrepancies with Claude Haiku 4.5 and GPT-5 mini. The top left bars show the same numbers as Figure 7, conservatively calculated as the fraction of patches that are flagged by both models. The remaining charts show annotation results for specific categories. Because both models make somewhat different choices in how they categorize the same functional discrepancies, there is somewhat less agreement per category than for the overall assessment.

# B. Extended Discussion

## B.1. Complementary Considerations

In addition to the four core dimensions discussed in §3, we recognize several complementary properties influencing the deployment of human-centered coding agents, though we argue they are best understood as emerging mechanisms rather than primary interaction primitives. Below, we highlight several considerations that came up during the process of putting together our position. Per area, we provide our brief definition, explain our reasoning for viewing it as complementary, and clarify how it relates to or emerges from the four core dimensions.

**Proactivity.** Proactive behavior refers to an agent's ability to anticipate user needs and act without explicit instruction. For instance, based on explicit user request or implicit observations across multiple sessions, an agent might automatically run a specific type of linter after code changes or user a certain color palette for future user interface components. While certainly useful for improving user experience and reducing friction, human-agent coding collaboration can function fully without proactivity: users can explicitly request every action, the agent responds, and the loop completes. It enhances convenience but is not required for the fundamental interaction to succeed.

Effective proactivity emerges naturally from *adaptability*. While simple proactive behaviors can be implemented via static rules (e.g., always run tests after code changes), calibrated proactivity requires knowing when such actions are welcome versus intrusive. An agent that learns a user's preferences, expertise level, and workflow patterns can determine when to take initiative versus wait for instruction (Horvitz, 1999). Without such learned context, proactive actions risk being unwelcome interruptions rather than helpful assistance.

**Safety.** Safety refers to preventing harmful, unintended, or irreversible agent behavior. For instance, an agent should not delete critical files without confirmation, execute unvetted external code, or make sweeping changes that corrupt a codebase. But this is also why we find that the definition of safety is difficult to formalize as a universal property, because its definition depends heavily on context. With regard to the given example, it could also be argued that a file deletions may be desirable in one workflow (automated cleanup scripts), but catastrophic in another (production deployment).

We believe that safety failures can best be thought of as a product of breakdowns in existing pillars rather than the absence of a standalone safety mechanism. Failures may arise from multiple interacting gaps: a file deletion gone wrong due to insufficient *steerability* (no checkpoint before an irreversible action), while a corrupted codebase may reflect insufficient *verification* (output not assessed before commit) (Leveson, 2016). When these pillars function well, agents that behave respectfully with respect to human expectations emerges as a natural, desirable side effect.

**Orchestration.** Orchestration refers to the coordination of multiple agents and humans working together on shared or interdependent tasks. For instance, a team of developers might delegate subtasks to different specialized agents, or multiple agents might collaborate on a large refactoring effort while a human architect provides high-level oversight.

Our position focuses on the fundamental dyad of a single human and a single coding agent. While we agree that orchestration's potential and rich additional challenges (delegation, conflict resolution, group/pluralistic alignment) are worth investigating, it is likely that successful orchestration should be founded upon the pillars for our basic, dyadic setting functioning well. Just as distributed systems build upon well-defined node-level protocols, we view orchestration as extending the interaction primitives discussed in this work to multi-party settings. Effective multi-party collaboration requires that each dyad achieves task alignment, that handoffs preserve steerability, that distributed outputs remain verifiable, and that agents adapt to broader team conventions.

We leave an extra note that there are certainly several works for multi-agent coordination for coding and software engineering (Hong et al., 2024; He et al., 2025). However, prior works have overwhelmingly explored collaboration settings that don't involve human participation.

**Customizability.** Customizability refers to a user's ability to manually configure agent behavior through skills, plugins, hooks, rules files, or other extensions. For instance, users might add custom linting rules, define project-specific commands, or install plugins that integrate with their preferred toolchain.

Customizability is a helpful system feature that achieves outcomes similar to adaptability, but through explicit user effort rather than agent learning. The core interaction loop functions without user-defined extensions; they enhance the experience but do not enable it. Note that customizability differs from *steerability* in several manners, most notably in temporal scope. Steerability concerns in-the-moment control during task execution, while customizability involves adding persistent

configuration that spans sessions. In this sense, customizability and *adaptability* are complementary: customizability provides immediate, user-controlled adjustments, while adaptability enables agents to internalize patterns over time without repeated manual configuration.

**Usability.** Usability concerns how easily users can interact with an agent system, including interface clarity, interaction friction, and opportunities for learning. For instance, whether a coding agent displays diffs inline or in a separate panel is a usability decision. Although essential for real-world adoption, these factors are largely determined by design choices rather than underlying interaction capabilities, and we therefore view usability as complementary to the core pillars.

### B.2. On the Horizon

There is a potential disillusionment that readers may feel about this work. Are our proposals stepping stones to a more collaboration-oriented future? Or are we simply conceptualizing listening devices to elicit human signals for improving fundamentally autonomous models? The second interpretation implies that our framework could accelerate knowledge worker displacement, reducing humans to transient reward models for systems designed to eventually replace them.

The truth is, we don't know. Our argument is narrower: current research optimizes for full autonomy prematurely before we understand what effective collaboration looks like. Whether full autonomy is the end state and how human-AI labor markets are affected en route are important but separate questions we aren't empirically equipped to answer. That said, our opinion is that human involvement is not a stopgap, but intrinsically necessary. For accountability, for judgment in novel situations, for alignment with intent only humans can ultimately adjudicate.