# OpenReview forum: "Position: Humans are Missing from AI Coding Agent Research"
_ICML.cc/2026/Position_Paper_Track — Submitted to ICML 2026 Position Paper Track_

### Official Review · Reviewer_ZTUV · 2026-03-10

**Significance:** 4
**Argument Clarity:** 4
**Rating:** 5
**Confidence:** 5

**Questions:**

Please see strengths and weaknesses, above.

**Alternative Views Section:**

Yes

**Compliance With Llm Reviewing Policy A Conservative:**

Affirmed.

**Discussion Potential:**

4

**Final Justification:**

My concerns were addressed, and I maintain my score.

**Paper Summary:**

The authors argue that development of coding agents thus far has focused almost exclusively on autonomous task completion; existing benchmarks measure the ability of coding agents to complete tasks without human supervision.  However, real-world use of coding agents necessarily involves human supervision -- humans must at least specify the problem and accept or reject the results, and miscommunication between human and agent causes a great deal of user frustration.

The authors identify four areas where communication between humans and agent could be improved.

1. Task Alignment: both human and agent must have a shared understanding of the task.
2. Steerability: the agent must continue to respond to human direction as the task is being done.
3. Verification: the artifacts produced by the agent must be easy for the human to review.
4. Adaptability: the agent must be able to learn over time and adapt to human preferences.

The authors argue that researchers should design agents, and develop benchmarks, that involve humans in the loop.  There are two ways to do so scalably:

  * create simulated humans with LLMs.
  * collect real human data from deployed coding tools, such as cursor.

**Position:**

Yes

**Position In Title:**

Yes

**Related Work:**

4

**Strengths And Weaknesses:**

The paper is well-written, has extensive references, and an excellent “alternative views” section.  The position is clearly stated, and the authors’ arguments are well-reasoned and compelling.  Although the authors limit their arguments to coding agents in particular, this line of reasoning easily generalizes to AI alignment more broadly, which is a critically important topic of immediate concern.  I recommend publication.

As far as "inspiring discussion" is concerned, the authors have identified a clear limitation of current coding agents, which means that their critique is both timely and relevant.  I certainly hope that the developers for those agents read this paper.  :-)  The authors also outline some strategies that researchers could potentially use to make AI coding tools more developer-centric.

That being said, I feel that the authors missed an opportunity to make two very important points.

First, wrt. Task Alignment:  it’s important to explain why alignment is hard.  The fundamental reason is that natural language is inherently ambiguous -- it is open to interpretation in numerous ways, and a great deal of information goes unsaid.  Please reference some work on formal program specification.  The lesson that the formal methods community learned a long time ago is that the only truly precise specification of program behavior is the code itself.  It is simply not possible to precisely describe program behavior in natural language, nor is it possible in general to provide a specification in a formal language that is less verbose than the code itself.  Thus, no matter how good agents become, an NL specification will never be complete, and will always have ambiguity and multiple interpretations.  Task alignment is intrinsically unsolvable without dialogue between human and agent.

Second, wrt. to both Task Alignment and Steerability:  The most important thing a coding agent can do is to (1) identify potential areas of uncertainty, and ***(2) ask clarifying questions!***  Speaking as a frustrated user here, I find that the most effective way to use current tools is  to keep an eagle eye on the chain-of-thought trace as it streams by, with my mouse hovering over the “stop” button.  For just about any real-world task, the agent will eventually hit an issue that I didn’t explain clearly enough.  My only recourse is to figure out when it’s going off the rails by reading the CoT log, hit “stop”, type in a clarification, and hit “run” again.  Doing so is much more efficient than having to reject code and redo the prompt, but it requires white-knuckle supervision -- definitely not an ideal user experience.

What the agent should do is identify the problem, stop on its own, and then ***ask me how I want to proceed.***  In other words, there should be a dialogue between human and agent, in which implementation details are nailed down, before a single line of code is produced.  This is a much cleaner way of managing the “clarification spiral” that the authors describe.

One possible reference here is: https://arxiv.org/abs/2402.03271, (I am not an author), but there may be others.

A minor nit is that I found the "adaptability" argument to be the weakest of the four.  LLMs these days are often quite good at remembering information; the problem is getting them to forget it.  Speaking anecdotally, I find myself frequently having to clear chat history just to keep the models from hauling in irrelevant previous context.  Having human control over what LLMs remember is arguably more important than adaptability.

**Support:**

3

---

> ### Author Rebuttal · Authors · 2026-03-31
>
> We thank reviewer `ZTUV` for their thorough and deeply insightful review.
> We are grateful for the reviewer’s strong endorsement of our position, and particularly value the substantive suggestions drawn from not only more formal research discussions, but also firsthand experience as a coding agent user.
> We resonate with these perspectives, and found the feedback constructive towards strengthening our paper.
>
> ---
>
> **\> W1: Need to explain why alignment is hard**
>
> We dedicated the Motivation section (line 145+) to this question, focusing on ambiguity in NL user queries as a key source of misalignment.
>
> That said, the reviewer's insight on how findings in formal methods literature connect to our arguments is very useful, as it adds a theoretical dimension to our argument.
> The inherent incompleteness of natural language specifications that contrasts sharply with the precision of program behavior is a very principled basis for explaining why one shot prompting is not enough for task alignment.
> We thank the reviewer for such rich insights and plan to incorporate this angle and relevant references in the revised version.
>
> ---
>
> **\> W2: Agents should identify uncertainty and ask clarifying questions. The "white-knuckle supervision" experience is not ideal.**
>
> We agree with the reviewer's perspective, which we directly discuss in Sections 3.1 and 3.2.
> The reviewer's description of monitoring chain-of-thought while hovering over the stop button is a vivid illustration of the steerability gap discussed in Section 3.2.
> Thanks for such a neat, clear example.
> We will incorporate both these ideas and the suggested reference in Sections 3.1 and 3.2 of the revision.
>
> We also offer a complementary view: While the reviewer's scenario envisions a *proactive* agent that queries the user before proceeding, the *adaptability* dimension (Section 3.4) can achieve similar goals (i.e., fulfill the user's request properly) by implicitly modeling the user and adjusting their behaviors accordingly, without being overly proactive or intrusive.
> In other words, such modeling could reduce the need for explicit interruption without trading off on alignment.
> An agent that is in tune with a user's preferences and conventions could resolve ambiguities silently.
> This connection may also address the reviewer's question in W3.
>
> ---
>
> **\> W3. Adaptability may be less important than human control in agent memory.**
>
> We found the reviewer's concern about how memory accumulation could actively degrade an agent's utility to be practical and well-founded.
> Forgetting is just as important as remembering.
> This resonates strongly with our adaptability discussion, where we argue how agents should strike a balance between learning and overfitting to stale or irrelevant context.
>
> As discussed in W2, we think adaptability encompasses both modeling of user preferences and corresponding adjustments in agent behavior, making it foundational to continual improvement across the other three dimensions (task alignment, verifiability, steerability).
> We also agree that preserving human control in agent memory is important to ensure desired properties we presented.
> We will integrate this point into our discussion and thank the reviewer for helping us construct a more comprehensive discussion.

---

> > ### Author Rebuttal · Reviewer_ZTUV · 2026-04-03
> >
> > My comments were addressed, and I maintain my score of "accept."

---

> > > ### Author Response · Authors · 2026-04-05
> > >
> > > Thanks again for your thoughtful review of our paper, we really appreciate it!

---

### Official Review · Reviewer_3gE7 · 2026-03-11

**Significance:** 3
**Argument Clarity:** 3
**Rating:** 5
**Confidence:** 2

**Questions:**

1. The paper introduces four key dimensions (task alignment, steerability, verifiability, and adaptability) to characterize human–agent collaboration in coding systems. Could the authors clarify how these dimensions could be operationalized in concrete evaluation protocols? For example, what measurable metrics or benchmark designs would allow researchers to quantify improvements along each of these axes?

2. The paper notes that verification and interaction requirements may differ across users with different levels of programming expertise. Do the authors think of what kind of evaluation frameworks explicitly account for user skill diversity, and if so, how might such evaluations be structured?

**Alternative Views Section:**

Yes

**Compliance With Llm Reviewing Policy A Conservative:**

Affirmed.

**Discussion Potential:**

3

**Final Justification:**

Overall, I think this is a valuable topic and would like to maintain my score. The rebuttal reinforces my prior assessment.

**Paper Summary:**

The paper states that current research on AI coding agents focuses on autonomous task completion and benchmark performance, while not sufficiently focusing on the role of the human developer and user in the interaction with the agent. The authors observe that many recent systems focus on improving success rates on benchmarks such as software engineering tasks, emphasizing increasingly capable autonomous agents and complex orchestration frameworks. However, these evaluation settings often overlook how humans specify tasks, guide the agent during execution, verify the produced solutions, and develop long-term collaboration with the system.

**Position:**

Yes

**Position In Title:**

Yes

**Related Work:**

3

**Strengths And Weaknesses:**

**Strength**: The paper addresses a timely topic at the intersection of machine learning, software engineering, and human–computer interaction. While recent work on coding agents and autonomous software development systems largely emphasizes benchmark performance and task autonomy, the paper argues that such evaluation overlooks the human role in task definition, intermediate guidance, output verification, and long-term interaction with coding systems. This perspective highlights a gap between benchmark-driven progress and real-world deployment, which is increasingly relevant as machine learning systems operate in collaborative environments.
Another strength is its positioning within current coding-agent research. The paper references benchmark-driven advances in software engineering tasks and autonomous agent frameworks, noting that improvements in benchmark scores do not necessarily reduce human effort in real development workflows. In particular, the discussion of verification cost and user oversight suggests that higher task completion rates may still impose significant cognitive and validation burdens on users. These observations motivate the need for interaction-centered evaluation.

**Weakness**: Although the paper presents a clear position, its arguments are primarily conceptual and rely on qualitative observations about existing benchmarks and development workflows rather than empirical evidence. While the paper references several benchmarks and agent systems, the related work could be expanded to more deeply incorporate literature from human–computer interaction, and collaborative programming environments, which would help further ground and strengthen the paper’s argument.

**Support:**

3

---

> ### Author Rebuttal · Authors · 2026-03-31
>
> We thank reviewer `3gE7` for recognizing the timeliness and importance of our work.
> The reviewer's observation that benchmark scores don't necessarily correlate with a reduction in human effort in real-world software development workflows is central to our argument.
>
> ---
>
> **\> W1a: Arguments are conceptual/qualitative instead of empirical.**
>
> We acknowledge that position papers are inherently conceptual by nature, as the contribution lies in synthesizing observations and proposing research directions rather than delivering experimental results.
> That said, our position is empirically grounded:
>
> * We conducted a large-scale analysis of top-performing agents on SWE-bench Verified (one of the most representative of the field), and use the patch bloat as a concrete measure to demonstrate the lack of user-facing utility of coding agents, and hence the necessity of focusing more on human-centered coding agents.
> * We also present a functional discrepancy analysis in the Appendix showing that over 50% of resolved SWE-bench patches differ functionally from gold solutions despite passing all unit tests.
>
> ---
>
> **\> W1b: Related work could add more HCI literature.**
>
> We reference relevant HCI studies throughout Sections 2 and 3, and situate our proposed research directions within existing HCI progress (Section 4).
> That said, we are happy to expand coverage in the revision and welcome any specific suggestions from the reviewer.
> For instance, we could explicitly connect to work on collaborative programming environments and mixed-initiative interaction design.
>
> ---
>
> **\> Q1: How could the four dimensions be operationalized in concrete evaluation protocols?**
>
> According to our formulation in Section 3, concrete evaluation metrics for each dimension are:
>
> 1. **Task alignment:** similarity between the internal intent representation of human and agent;
> 2. **Steerability:** (a) accuracy of the agent in identifying control points during task-solving, and (b) similarity between agent trajectories under real human intervention versus an ideal trajectory;
> 3. **Verification:** accuracy and efficiency of human verifying intermediate and final agent output;
> 4. **Adaptability:** improvement of task performance (e.g., success rate) across a certain number of tasks, given a fixed user/task distribution.
>
> We acknowledge that some representations are hard to capture due to their task/human/agent-dependent nature. We intend our work as a conceptual foundation, and welcome future work to propose approximations suited to particular setup.
>
> ---
>
> **\> Q2: What evaluation frameworks could account for user skill diversity?**
>
> As we illustrated in Figure 4, adapting the evaluation medium is one way to account for diverse user skill levels. For instance, users with limited programming knowledge may respond better to visual output or easily accessible ways of verifying behavior interactively rather than to raw source code, allowing verification without programming proficiency prerequisites.

---

> > ### Author Rebuttal · Reviewer_3gE7 · 2026-04-03
> >
> > Thanks to the author for the response. I would maintain my recommendation of acceptance.

---

> > > ### Author Response · Authors · 2026-04-05
> > >
> > > Thank you once again for your review of our paper!

---

### Official Review · Reviewer_RBFP · 2026-03-12

**Significance:** 2
**Argument Clarity:** 3
**Rating:** 1
**Confidence:** 4

**Questions:**

line 42: "Much of the field has focused on advancing autonomy, measured by success rate on harder benchmarks." Could you also provide examples of research that do not rely solely on success rates but still have limitations in reflecting human interactions?

line 47: "In this setting, the goal of developing coding agents is not to replace human labor, but to augment complementary human strengths such as initiative and judgment." How this is concluded from "Instead, it unfolds through iterative interaction, partial delegation, evolving goals, and continuous human oversight." Namely, why the need for a coding agent from human interaction (i.e., "iterative interaction, partial delegation, evolving goals, and continuous human oversight") can equal a human's need (i.e., "augment complementary human."). They are not even the same subjects. If humans do not need this, why do we set the goal of developing coding agents as "to augment complementary ...".

line 80: "We choose these dimensions as interaction primitives that together span the agent task-solving cycle". Did you derive this cycle from any existing cycles or build a house on sand?

Your definitions are not grounded. For example,
line 152: "Definition. Task alignment refers to the process by which humans and agents establish and maintain a shared task understanding through mutual modeling (Clark & Brennan, 1991)."
First, this referred paper is talking about "Grounding" in "Communication", where both definitions, "Grounding" and "Communication", have been defined. However, I didn't see a connection between that paper to this defined Task Alignment. Second, given that this theory is connected with Task Alignment, "Communication" needs to be adapted to the Agent-Human content, which has been missed.

Similarly,
line 192: "Definition. Steerability concerns an agent’s ability to expose and respond to human control signals throughout task execution. Rather than optimizing solely for uninterrupted autonomy (Horvitz, 1999), ...."
I cannot find any definition of "Steerability" in the referred paper.

**Alternative Views Section:**

Yes

**Compliance With Llm Reviewing Policy A Conservative:**

Affirmed.

**Discussion Potential:**

1

**Final Justification:**

The rebuttal does not address my main concerns. I maintain my prior evaluation.

**Paper Summary:**

This paper advocates that the research on AI coding agents should shift from being autonomous to human-centered. Accordingly, a new research direction is proposed: human-centered coding agents. Instead of improving on the capability of task completion, the next-step research in this domain should focus on whether a human can understand, trust, and work with it. Motivated by that, this paper proposes a human-coding agent collaboration loop that aims to unify the human interaction stages during the coding agents executing a task. For better quantification, this paper defines four pillars and formalizes a computable metric for each.

**Position:**

Yes

**Position In Title:**

Yes

**Related Work:**

1

**Strengths And Weaknesses:**

Strengths:
It is an emerging topic, and the idea has been clearly presented.

Weaknesses:
1. This topic involves an interdisciplinary understanding of computer science, software engineering, and human-centered interactions, as it discusses one computer science technique applied to a software engineering problem that requires human interaction. However, this paper fails to draw a picture that fully covers the necessary background from at least these three domains.

2. I assume this paper to be positioned in the computer science community based on its argument. However, it fails to clarify the boundary of computer science researchers from the other two.

3. There are major issues with research rigor, e.g., claims are not well-supported but are used directly as a conclusion; Theories are either wrongly introduced or not adapted before use.

**Support:**

1

---

> ### Author Rebuttal · Authors · 2026-03-31
>
> We thank reviewer `RBFP` for their recognition of the potential importance of our position, and the clarity of our discussion.
>
> **\> W1: Lack of necessary background from CS, SE, and HCI.**
>
> We survey coding agents across CS and SE communities in Section 2, and integrate HCI studies throughout Section 3 and Section 4.
> We appreciate the reviewer's attention to interdisciplinary grounding and welcome specific pointers so we can address gaps in our revision.
>
> **\> W2: Unclear boundary of CS w.r.t. SE and HCI.**
>
> Our position is grounded in the ML/NLP community, with a major focus on building better coding agents. This naturally overlaps SE as the field increasingly adopts neural-based approaches. Throughout our paper, we integrate HCI insights that study coding agents in their relationship to humans. Nonetheless, we note that ML/NLP, SE, and HCI are all subfields within the broader CS community with different emphases. We will clarify this framing in the revision to avoid confusion about interdisciplinary boundaries.
>
> **\> W3: Issues in research rigor**
>
> We invested substantial effort in grounding our position through a comprehensive literature review (e.g., Section 2) and concept formulation (Section 3).
> We address specific examples in our responses below, and welcome further pointers to any additional instances.
>
> **\> Q (line 42): Examples of research beyond success rates that still lack human interaction?**
>
> While most work uses unit tests to measure task completion, some research also evaluates coding systems on dimensions such as program efficiency [1], code quality and verbosity [2], and long-term maintainability [3]. Yet in all of these examples, the agent operates fully autonomously with no human (or human simulator) in the loop. Our claim is not about which metric is best for measuring coding systems, but that the systems under study are fundamentally end-to-end autonomous, leaving no room to study how humans interact with, steer, or verify agent behavior. We will clarify that our broader claims hold for a more diverse set of metrics in the revision.
>
> * [1] Waghjale et al. “ECCO: Can we improve model-generated code efficiency without sacrificing functional correctness?”
> * [2] Orlanski et al. “SlopCodeBench: Benchmarking How Coding Agents Degrade Over Long-Horizon Iterative Tasks”
> * [3] Chen et al. “SWE-CI: Evaluating Agent Capabilities in Maintaining Codebases via Continuous Integration”
>
> **\> Q (line 47): Logical connection between “iterative interaction, partial delegation...” and “augment complementary human strengths”?**
>
> The core question is why the collaborative nature of real-world coding implies agents should augment rather than replace human capabilities. Our argument is that the interactions we describe (partial delegation, evolving goals, continuous oversight) are not just features of how work currently happens to be organized; they require human initiative and judgment grounded in context, preferences, and domain knowledge the agent does not have independent access to. Because these interactions inherently require human input, the agent's role is necessarily complementary.
>
> **\> Q (line 80): Derivation of the agent task-solving cycle?**
>
> This task-solving cycle is grounded in foundational agent definitions [6] and consistent with the broader agent-building literature.
>
> [6] Stuart J Russell and Peter Norvig. Artificial intelligence a modern approach. London, 2010.
>
> **\> Q (line 152): Connection between Clark & Brennan (1991) and Task Alignment?**
>
> Clark & Brennan (1991) define “grounding” as the process by which participants in communication establish mutual understanding. Our concept of “task alignment” applies this same principle to the specific setting of human-agent coding collaboration: the human and agent must converge on a shared understanding of the task. The key idea we draw from Clark & Brennan is that both parties actively work to maintain common ground throughout the interaction.
> We agree with the reviewer that this transfer should be made more explicit, and will clarify it in the revision.
>
> **\> Q (line 192): Horvitz (1999) does not define “Steerability.”**
>
> To clarify, the first sentence presents our own proposed definition of steerability, independent of prior work. The following sentence offers broader context around this concept, where we draw on insights from Horvitz (1999), to situate our definition within the relevant literature. Specifically, a central argument of Horvitz (1999) is that intelligent agents should not act with uninterrupted autonomy; instead, the paper proposes 12 principles for mixed-initiative user interfaces, several of which directly inform our notion of steerability, e.g., employing dialog to resolve uncertainties about user goals (principle 5) and providing mechanisms for agent-user collaboration to refine results (principle 9). We appreciate the reviewer's catch, and will revise our phrasing to make the citation's role clearer.

---

> > ### Author Rebuttal · Reviewer_RBFP · 2026-04-05
> >
> > > W1: This position paper would like to argue for a "human-centered coding agent". However, the reviewed references focus on coding agents only. There is a lack of review on how CS/SE/HCI acknowledges the human involvement in coding agents. For example,
> > [r1] https://arxiv.org/abs/2512.23844
> > [r2] https://dl.acm.org/doi/full/10.1145/3708359.3712089
> > I couldn't find any in your reference that explicitly contains "human-centered" + "agent". Could you list the reference of "human-centered coding agent" rather than "agent" only?
> >
> > > W2: Clearly, [r1,r2] have discussed human-centered agents in their domain. Given this paper targets the ML/NLP community, how does the conclusion benefit the ML/NLP community in a way that is different from/similar to the other communities? As you didn't review similar arguments in other communities while positioning your argument as an interdisciplinary understanding of computer science, software engineering, and human-centered interactions, how could I understand the contribution as uniquely benefiting the ML/NLP community rather than reiterating existing conclusions that are highly acknowledged in other communities?
> >
> > > W3&Q152&Q192: As you have mentioned, Clark & Brennan (1991) define “grounding” rather than “task alignment”. Similar to the definition of "Steerability". These definitions form the basis for understanding your conceptual model. Given that their adaptability is not well-explained, I couldn't assess the rigor of the proposed position.
> >
> > > Q (line 42): addressed
> >
> > > Q (line 47): My question is not addressed, so I would like to ask it in another way. Let's assume we accept the argument that a human-centered coding agent is preferred over an autonomous agent. My question is whether this "preference" is from human perception or system-level considerations?
> > If it is from the human side, please refer to my previous question, "If humans do not need this, why do we set the goal of developing coding agents as "to augment complementary ..."."
> > If it is from system-level considerations, could you specify which system-level considerations you are referring to?
> >
> > > Q (line 80): I didn't find [6] in the paper, so I assume this is a newly added reference. Please correct me if I missed it in the main paper or the Appendix. In addition, Ref [6] seems to be a book; could you specify the chapter?
> > [6] Stuart J Russell and Peter Norvig. Artificial intelligence a modern approach. London, 2010.

---

> > > ### Author Response · Authors · 2026-04-08
> > >
> > > We have substantively engaged with all concerns across two rounds. Where we agree, we commit to concrete revisions; where we disagree, we have provided evidence and reasoning in our detailed responses below. We believe the remaining disagreements reflect differences in expectations for position papers rather than identified gaps.
> > >
> > > **\> W1. Lack of review on how CS/SE/HCI acknowledges human involvement.**
> > > We argue that we have broad coverage of literature on human involvement with coding agents; references discussing humans' interactions with coding agents appear at: Line 470 (left), 491 (left), 458 (right), 463 (right), 468 (right), 502 (left), 545 (left), 522 (right), 553 (left), 577 (left), 582 (left), 554 (right), 560 (right), 573 (right), 578 (right), 626 (left), 639 (left), 652 (right), 668 (left), 673 (left), 697 (right).
> > > We thank the reviewer for the additional references. [r1] offers a qualitative taxonomy of coding agent behaviors from user interviews; [r2] addresses conversational AI in group brainstorming, a different domain. While valuable, these works illustrate the gap our paper addresses: existing contributions remain either qualitative or outside the coding agent setting. Our paper provides a formalized framework for quantitative evaluations of human-centered coding agents in the ML/NLP community. We are happy to include both references in our revision.
> > >
> > > **\>W2. How does the conclusion benefit the ML/NLP community.**
> > > As we stated in our opening position (page 1, top right), our paper benefits the ML/NLP community by (i) revealing the under-emphasized human aspect in building and training coding agents, and (ii) providing concrete problem formulations (Sec 3) and actionable research directions (Sec 4). We would like to emphasize that the ML/NLP community is where coding agents are actually built: architectures are designed, models are trained, and scaffolds are engineered. Even where HCI has identified relevant user needs qualitatively, translating those insights into computable metrics and technical research agendas for the ML community is a distinct and necessary contribution. Indeed, as our survey in Sec 2 shows, the ML/NLP community has not yet internalized these insights; the field remains converged on autonomy-first benchmarks. For HCI readers, our contribution is complementary: moving beyond qualitative case studies (such as [r1]) toward rigorous mathematical formulations and scalable evaluation infrastructure. We are happy to make these contributions more prominent in the revision.
> > >
> > > **\>W3/etc. Reviewer expressed “I couldn’t assess the rigor of the proposed position”**
> > > We respectfully note that our four pillars are *definitions* that we propose as a contribution, not claims that require derivation from prior work. The citations (e.g., Clark & Brennan 1991, Horvitz 1999) serve to situate our concepts within established theoretical foundations, not to derive them: grounding in communication (Clark & Brennan) inspires task alignment; mixed-initiative interaction principles (Horvitz) motivate steerability. We will make these relationships more explicit in revision. Our definitions closely follow the empirical evidence presented in Sec 3 (developer surveys, industry reports, user studies). If the reviewer questions whether these definitions are useful or serve the goals outlined in the paper, we are happy to clarify further.
> > >
> > > **\>Q (line 47). Is the preference over human-centered coding agents from human perception or system-level considerations?**
> > > Our preference for human-centeredness comes from human perception (see our position on page 1, top-right). Coding agents (Claude Code, Cursor, GitHub Copilot) are already widely used by humans. Throughout Sec 3, we present evidence from developer surveys, industry reports, and user studies showing that practitioners struggle with communicating intent, verifying outputs, and steering agent behavior; these directly motivate our formalizations.
> > >
> > > Regarding system-level evidence: current benchmarks overwhelmingly evaluate autonomous task completion with no human in the loop, making it all but impossible to measure whether human involvement improves system-level outcomes. Rather than undermining our position, this gap is a core part of what we argue needs to change. Our proposed formalizations (Sec 3) and research directions (Sec 4) are designed precisely to make such system-level evaluation possible.
> > >
> > > **\>Q (line 80). Reference and specific agent definition in [6].**
> > > The accurate reference should be: Russell, Stuart, Peter Norvig, and Artificial Intelligence. "A modern approach." Artificial Intelligence. Prentice-Hall, Egnlewood Cliffs 25.27 (1995): 79-80.
> > > The specific location of the agent definition is in: Chapter 2.1 "Agents and Environments", first sentence.
> > > Thanks for the reviewer’s suggestion, we are happy to incorporate this reference to our discussion.

---

### Official Review · Reviewer_bZQf · 2026-03-13

**Significance:** 3
**Argument Clarity:** 3
**Rating:** 4
**Confidence:** 4

**Questions:**

On Steerability: What specific methods do you suggest for making coding agents more "steerable" when dealing with complex, multi-step projects that require iterative human intervention?

On Adaptability: Can you provide concrete examples or scenarios in which adaptability in coding agents—such as maintaining a memory of user preferences—would be most beneficial, particularly in high-stakes tasks?

On Verification: What are your thoughts on the limitations of current verification methods (such as unit testing) for more advanced, general-purpose coding agents, and how do you envision improving the verification process for non-technical users?

**Alternative Views Section:**

Yes

**Compliance With Llm Reviewing Policy A Conservative:**

Affirmed.

**Discussion Potential:**

3

**Final Justification:**

The author solved my problem, I'll keep my score, and suggest borderline accept.

**Paper Summary:**

This paper argues for a shift in AI coding agent research, which has primarily focused on autonomy and task completion, toward a human-centered approach. It emphasizes the importance of improving collaboration between human developers and coding agents, not just enhancing the agent’s solo task-solving capabilities. The paper proposes research directions that include designing user-involved coding environments, improving verification mechanisms, and creating metrics for human-agent interaction quality.

**Position:**

Yes

**Position In Title:**

Yes

**Related Work:**

3

**Strengths And Weaknesses:**

**Strengths**

- This paper presents a timely argument in light of the increasing use of AI in coding.

- The authors take a clear stance that current research in AI coding agents should pivot towards human-centered models. The well-defined four pillars—task alignment, steerability, verification, and adaptability—serve as a concrete foundation for this shift.

- The paper effectively outlines actionable research directions. By calling for the development of user simulators and scalable human modeling, the authors provide forward-thinking ways to address existing gaps in human-agent collaboration.

**Weaknesses**

- The proposed human-centered design approach, while theoretically sound, lacks a detailed discussion of the practical challenges involved in its implementation. The paper could have explored the technical complexities of integrating human-centered design principles into existing agent frameworks.

**Support:**

3

---

> ### Author Rebuttal · Authors · 2026-03-31
>
> We thank Reviewer `bZQf` for acknowledging our argument's timeliness and the four pillars as a concrete foundation for actionable future research directions.
>
> ---
>
> **\> W1: Human-centered design discussion lacks practical implementation challenges.**
>
> We appreciate the suggestion and agree with the necessity of discussing implementation challenges, this is exactly why we formally defined metrics for each pillar in Section 3 despite the field being largely qualitative so far. We also list out several important challenges according to each technical direction forward:
>
> 1. **Scaling human modeling:** We discussed current limitations in user simulators, such as homogeneity in "preferences, expertise, and failure modes", as well as challenges such as scalability and data privacy issues when it comes to collecting real human data.
> 2. **Efficient oversight:** We discussed two key limitations of unit testing as the dominant verification paradigm. First, tests often capture only a narrow notion of functional correctness. Second, tests do not reduce verification effort: when the agent generates both code and tests, users must verify that tests capture their intent, a lateral shift rather than a reduction in cognitive load, which also presupposes programming fluency, making it ill-suited for non-technical users.
> 3. **Measuring human-agent interaction:** We point out the challenge of heavily contextualized metrics and architectures to unify and compare across systems at scale.
> 4. **Integration with existing capability-focused pipelines:** We discuss the challenge of incorporating human-centered objectives into current training and evaluation workflows (Section 5c), noting that RLHF serves as a precedent where incorporating human preferences into the training objective, rather than deferring them to downstream product work, led to a step change in practical usefulness (GPT-3 to ChatGPT).
>
> Our current framing may emphasize these as research opportunities rather than explicit challenges. In the revision, we are happy to restructure these sections to more clearly delineate: (a) the practical implementation barriers, (b) why they are difficult, and (c) our proposed paths forward.
>
> ---
>
> *Regarding questions*
>
> Our paper's primary goal is to establish the position that coding agent research should prioritize human-centered interaction over solo autonomous task completion.
> Within the allotted space, we attempt to provide concrete examples to the best of our ability.
> We view more detailed technical solutions as a natural, exciting direction for follow up work.
>
> ---
>
> **\> Q1. Specific methods suggestion in agent steerability**
>
> Consider using a coding agent to build a medical records dashboard for medical practitioners.
> Along the way, the agent would encounter design choices only the human can resolve:
>
> * Should patient data be displayed in full detail, or as redacted summaries?
> * Which roles should have access to which views?
> * When a data field is missing or malformed, should the dashboard show an error, display a placeholder, or silently skip the record?
>
> As described in the formulation (lines 202-215), each of these questions corresponds to a user control point in the problem-solving arc. One important aspect is how to measure these: this could be captured by the autonomy level of agents, or the granularity/frequency of human interaction. Training models to recognize such decision points in context, and further modeling user-preferred interaction patterns, could be one concrete path toward more steerable agents.
>
> ---
>
> **\> Q2. Concrete examples of needing adaptability**
>
> In many open-source settings, developers directly issue database queries (e.g., via raw SQL).
> However, often times a financial service company must respect certain security and privacy practices that then requires all database queries to go through an internal ORM layer to ensure compliance and traceability.
> The developer tells the agent.
> An adaptable agent would memorize and abide by the company's specific procedural requirement going forward, which (1) improves task correctness with respect to internal criteria, and (2) relieves developers from having to repeat themselves.
> An agent that doesn't adapt forgets this constraint in subsequent sessions.
> Completely unbeknownst to the developer, the agent then implements key features with direct SQL queries that lead to a data breach.
>
> ---
>
> **\> Q3. Limitations of verification methods**
>
> As unit tests are increasingly generated by AI, they impose a growing review burden on technical users while remaining inaccessible to non-programmers. As discussed in Figure 2 and lines 362-370, we think presentation is a key bottleneck. Alternative representations such as automatically generated screenshots for UI changes, control flow diagrams, interactive sandboxes, or executable notebooks could help users verify output without requiring programming proficiency. We hope our discussion opens up thoughts beyond "just test cases".

---

> > ### Author Rebuttal · Reviewer_bZQf · 2026-04-03
> >
> > My comments were addressed, and I maintain my score.

---

> > > ### Author Response · Authors · 2026-04-05
> > >
> > > We sincerely appreciate you taking time to review our work, thank you again!

---

### Decision · Program_Chairs · 2026-04-30

**Decision:**

Reject

**Comment:**

The majority of reviewers felt the paper stated its position clearly, situated its position within the current coding agent research, presented well-written arguments and backed up the arguments with extensive references. These reviewers also appreciated the authors' concrete recommendations to guide researchers towards achieving their vision. All reviewers agreed that the topic of AI coding agents is definitely timely and relevant.

In terms of weaknesses, reviewers wanted the authors to do a better job of situating the paper with relevant literature in other neglected areas such as collaborative programming environments and formal methods. A reviewer raised the issue that the paper should include a deeper discussion of why task alignment is challenging due to the ambiguity of natural language and the importance of having the agent ask clarifying questions. Finally, some reviewers wanted more clarity for terms used by the author. Overall, the paper could use a few more improvements to strengthen the position and situate it better with existing work.